# REINFORCEMENT LEARNING WITH ACTION SEQUENCE FOR DATA-EFFICIENT ROBOT LEARNING

## ABSTRACT

Training reinforcement learning (RL) agents on robotic tasks typically requires a large number of training samples. This is because training data often consists of noisy trajectories, whether from exploration or human-collected demonstrations, making it difficult to learn value functions that understand the effect of taking each action. On the other hand, recent behavior-cloning (BC) approaches have shown that predicting *a sequence of actions* enables policies to effectively approximate noisy, multi-modal distributions of expert demonstrations. Can we use a similar idea for improving RL on robotic tasks? In this paper, we introduce a novel RL algorithm that learns a critic network that outputs *Q-values over a sequence of actions*. By explicitly training the value functions to learn the consequence of executing a series of current and future actions, our algorithm allows for learning useful value functions from noisy trajectories. We study our algorithm across various setups with sparse and dense rewards, and with or without demonstrations, spanning mobile bi-manual manipulation, whole-body control, and tabletop manipulation tasks from BiGym, HumanoidBench, and RLBench. We find that, by learning the critic network with action sequences, our algorithm outperforms various RL and BC baselines, in particular on challenging humanoid control tasks.

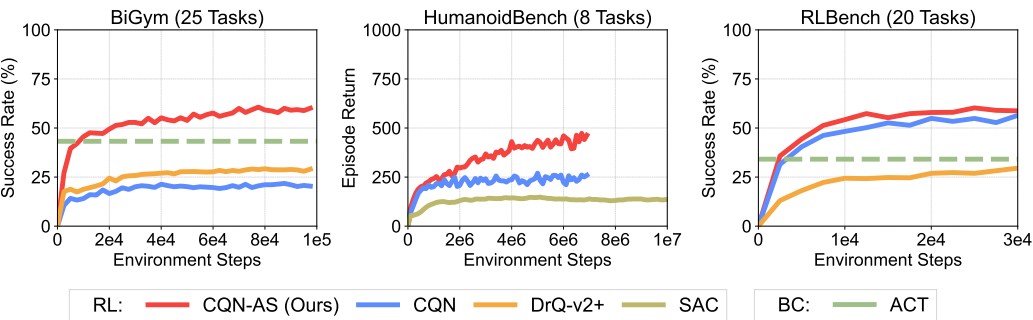

Figure 1: **Summary of results.** Coarse-to-fine Q-Network with **A**ction **S**equence (CQN-**AS**) is a value-based RL algorithm that learns a critic network with action sequence. We study CQN-**AS** on 53 robotic tasks from BiGym (Chernyadev et al., 2024), HumanoidBench (Sferrazza et al., 2024), and RLBench (James et al., 2020), where prior model-free RL algorithms struggle to achieve competitive performance. We show that CQN-**AS** outperforms various RL and BC baselines such as CQN (Seo et al., 2024), DrQ-v2+ (Yarats et al., 2022), SAC (Haarnoja et al., 2018), and ACT (Zhao et al., 2023).

## 1 INTRODUCTION

Reinforcement learning (RL) holds the promise of continually improving policies through online trial-and-error experiences (Sutton & Barto, 2018), making it an ideal choice for developing robots that can adapt to various environments. However, despite this promise, training RL agents on robotic tasks typically requires a prohibitively large number of training samples (Kalashnikov et al., 2018; Herzog et al., 2023), which becomes problematic as deploying robots often incurs a huge cost. Therefore many of the recent successful approaches on robot learning have been based on behavior-cloning (BC; Pomerleau 1988), which can learn strong policies from offline expert demonstrations (Brohan et al., 2023b;a; Zhao et al., 2023; Chi et al., 2023; Team et al., 2024; Fu et al., 2024a).

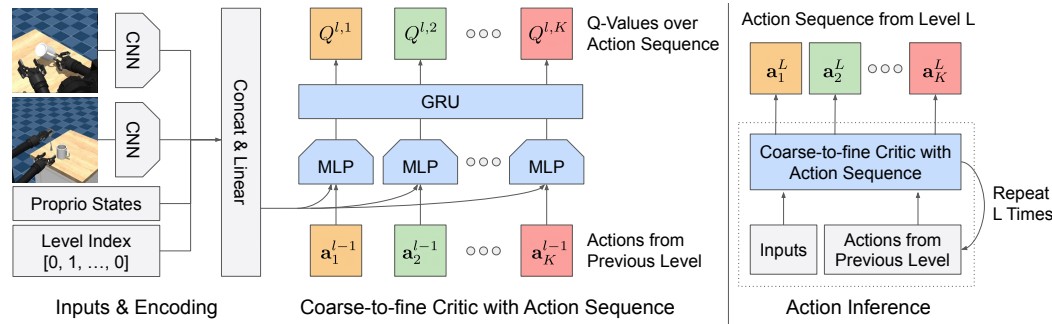

Figure 2: **Coarse-to-fine Q-network with action sequence.** (Left) Our key idea is to train a critic network to output Q-values over *a sequence of actions*. We design our architecture to first obtain features for each sequence step and aggregate features from multiple sequence steps with a recurrent network. We then project these outputs into Q-values at level $l$. (Right) For action inference, we repeat the procedure of computing Q-values for level $l \in \{1, ..., L\}$. We then find the action sequence with the highest Q-values from the last level $L$, and use it for controlling robots at each time step.

One cause for the poor data-efficiency of RL algorithms on robotic tasks is that training data consists of *noisy* trajectories. When collecting data for training RL agents, we typically inject some noise into actions for exploration (Sehnke et al., 2010; Lillicrap et al., 2016) that may induce trajectories with jerky motions. Moreover, we often initialize training with human-collected demonstrations that can consist of noisy multi-modal trajectories (Chernyadev et al., 2024). Such noisy data distributions make it difficult to learn value functions that should understand the consequence of taking each action. We indeed find that prior RL algorithms perform much worse than the BC baseline on mobile bi-manual manipulation tasks with *human-collected* demonstrations when compared to a tabletop manipulation setup with *synthetic* demonstrations collected via motion-planning (see Figure 1).

On the other hand, recent BC approaches have shown that *predicting a sequence of actions* enables policies to effectively approximate the noisy, multi-modal distribution of expert demonstrations (Zhao et al., 2023; Chi et al., 2023). Inspired by this, in this paper, we investigate how to use a similar idea for improving the data-efficiency of RL algorithms on robotic tasks. In particular, we present a novel RL algorithm that learns a critic network that outputs *Q-values over a sequence of actions* (see Figure 2). By training the critic network to explicitly learn the consequence of taking a series of current and future actions, our algorithm enables the RL agents to effectively learn useful value functions from noisy trajectories. We build this algorithm upon a recent value-based RL algorithm that learns RL agents to zoom-into continuous action space in a coarse-to-fine manner (Seo et al., 2024), thus we refer to our algorithm as Coarse-to-fine Q-Network with **A**ction **S**equence (CQN-**AS**).

To evaluate the generality and capabilities of CQN-**AS**, we study CQN-**AS** on various setups with sparse and dense rewards, and with or without demonstrations. In BiGym benchmark (Chernyadev et al., 2024), which provides *human-collected* demonstrations for mobile bi-manual manipulation tasks, CQN-**AS** outperforms various model-free RL and BC baselines (Yarats et al., 2022; Zhao et al., 2023; Seo et al., 2024). Moreover, in HumanoidBench (Sferrazza et al., 2024), which consists of *densely-rewarded* humanoid control tasks, we show that CQN-**AS** can also be effective without demonstrations, outperforming prior model-free RL baselines (Haarnoja et al., 2018; Seo et al., 2024). Finally, in RLBench (James et al., 2020), which provides *synthetic* demonstrations generated via motion-planning, CQN-**AS** achieves similar performance as model-free RL and BC baselines on most tasks, but significantly better performance on several long-horizon manipulation tasks.

## 2 BACKGROUND

**Problem setup** We mainly consider a robotic control problem which we formulate as a partially observable Markov decision process (Kaelbling et al., 1998; Sutton & Barto, 2018). At each time step $t$, an RL agent encounters an observation $\mathbf{o}_t$, executes an action $\mathbf{a}_t$, receives a reward $r_{t+1}$, and encounters a new observation $\mathbf{o}_{t+1}$ from the environment. Because the observation $\mathbf{o}_t$ does not contain full information about the internal state of the environment, in this work, we use a stack of past observations as inputs to the RL agent by following the common practice in Mnih et al. (2015).

For simplicity, we omit the notation for these stacked observations. When the environment is fully observable, we simply use $\mathbf{o}_t$ as inputs. Our goal in this work is to train a policy $\pi$ that maximizes the expected sum of rewards through RL while using as few samples as possible, optionally with access to a modest amount of expert demonstrations collected either by motion-planners or by humans.

**Inputs and encoding**    Given visual observations $\mathbf{o}_t^v = \{\mathbf{o}_t^{v_1}, ..., \mathbf{o}_t^{v_M}\}$ from $M$ cameras, we encode each $\mathbf{o}_t^{v_i}$ using convolutional neural networks (CNN) into $\mathbf{h}_t^{v_i}$. We then process them through a series of linear layers to fuse them into $\mathbf{h}_t^v$. If low-dimensional observations $\mathbf{o}_t^{\texttt{low}}$ are available along with visual observations, we process them through a series of linear layers to obtain $\mathbf{h}_t^{\texttt{low}}$. We then use concatenated features $\mathbf{h}_t = [\mathbf{h}_t^v, \mathbf{h}_t^{\texttt{low}}]$ as inputs to the critic network. In domains without vision sensors, we simply use $\mathbf{o}_t^{\texttt{low}}$ as $\mathbf{h}_t$ without encoding the low-dimensional observations.

**Coarse-to-fine Q-Network**    Coarse-to-fine Q-Network (CQN; Seo et al. 2024) is a value-based RL algorithm for continuous control that trains RL agents to zoom-into the continuous action space in a coarse-to-fine manner. In particular, CQN iterates the procedures of (i) discretizing the continuous action space into multiple bins and (ii) selecting the bin with the highest Q-value to further discretize. This reformulates the continuous control problem as a multi-level discrete control problem, allowing for the use of ideas from sample-efficient value-based RL algorithms (Mnih et al., 2015; Silver et al., 2017; Schrittwieser et al., 2020), designed to be used with discrete actions, for continuous control.

Formally, let $\mathbf{a}_t^l$ be an action at level $l$ with $\mathbf{a}_t^0$ being the zero vector[1]. We then define the coarse-to-fine critic to consist of multiple Q-networks which compute Q-values for actions at each level $\mathbf{a}_t^l$, given the features $\mathbf{h}_t$ and actions from the previous level $\mathbf{a}_t^{l-1}$, as follows:

$$Q_\theta^l(\mathbf{h}_t, \mathbf{a}_t^l, \mathbf{a}_t^{l-1}) \quad \text{for} \quad l \in \{1, ..., L\} \tag{1}$$

We optimize each Q-network at level $l$ with the following objective:

$$\mathcal{L}^l = \left( Q_\theta^l(\mathbf{h}_t, \mathbf{a}_t^l, \mathbf{a}_t^{l-1}) - r_{t+1} - \gamma \max_{a'} Q_{\bar{\theta}}^l(\mathbf{h}_{t+1}, a', \pi^l(\mathbf{h}_{t+1})) \right), \tag{2}$$

where $\bar{\theta}$ are delayed parameters for a target network (Polyak & Juditsky, 1992) and $\pi^l$ is a policy that outputs the action $\mathbf{a}_t^l$ at each level $l$ via the inference steps with our critic, *i.e.,* $\pi^l(\mathbf{h}_t) = \mathbf{a}_t^l$. Specifically, to output actions at time step $t$ with the critic, CQN first initializes constants $a_t^{\texttt{low}}$ and $a_t^{\texttt{high}}$ with $-1$ and $1$. Then the following steps are repeated for $l \in \{1, ..., L\}$:

- Step 1 (Discretization): Discretize an interval $[a_t^{\texttt{low}}, a_t^{\texttt{high}}]$ into $B$ uniform intervals, and each of these intervals become an action space for $Q_\theta^l$

- Step 2 (Bin selection): Find a bin with the highest Q-value and set $a_t^l$ to the centroid of the bin.

- Step 3 (Zoom-in): Set $a_t^{\texttt{low}}$ and $a_t^{\texttt{high}}$ to the minimum and maximum of the selected bin, which intuitively can be seen as zooming-into each bin.

We then use the last level's action $\mathbf{a}_t^L$ as the action at time step $t$. For more details, including the inference procedure for computing Q-values, we refer readers to Appendix B.

## 3 COARSE-TO-FINE Q-NETWORK WITH ACTION SEQUENCE

We present Coarse-to-fine Q-Network with **A**ction **S**equence (CQN-**AS**), a value-based RL algorithm that learns a critic network that outputs Q-values for *a sequence of actions* $\mathbf{a}_{t:t+K} = \{\mathbf{a}_t, ..., \mathbf{a}_{t+K-1}\}$ for a given observation $\mathbf{o}_t$. Our main motivation for this design comes from one of the key ideas in recent behavior-cloning (BC) approaches, *i.e.,* predicting *action sequences*, which helps resolve ambiguity when approximating noisy, multi-modal distributions of expert demonstrations (Zhao et al., 2023; Chi et al., 2023). Similarly, by explicitly learning Q-values of both current and future actions from the given state, our approach aims to mitigate the challenge of learning Q-values with noisy trajectories from exploratory behaviors or human-collected demonstrations.

This section describes how we design our critic network with action sequence (see Section 3.1) and how we utilize action sequence outputs to control robots at each time step (see Section 3.2). The overview of our algorithm is available in Figure 2.

---

[1]For simplicity, we describe CQN and CQN-**AS** with a single-dimensional action in the main section. See Appendix B for full description with $N$-dimensional actions, which is straightforward but requires more indices.

## 3.1 Coarse-to-fine Critic with Action Sequence

Our key idea is to design a critic network to explicitly learn Q-values for current action and future actions from the current time step $t$, *i.e.,* $\{Q(\mathbf{o}_t, \mathbf{a}_t), Q(\mathbf{o}_t, \mathbf{a}_{t+1}), ..., Q(\mathbf{o}_t, \mathbf{a}_{t+K-1})\}$, to enable the critic to understand the consequence of executing a series of actions from the given state.

**Formulation and objective**   Let $\mathbf{a}_{t:t+K}^l = \{\mathbf{a}_t^l, ..., \mathbf{a}_{t+K-1}^l\}$ be an action sequence at level $l$ and $\mathbf{a}_{t:t+K}^0$ be a zero vector. We design our coarse-to-fine critic network to consist of multiple Q-networks that compute Q-values for each action at sequence step $k$ and level $l$:

$$Q_\theta^{l,k}(\mathbf{h}_t, \mathbf{a}_{t+k-1}^l, \mathbf{a}_{t:t+K}^{l-1}) \text{ for } l \in \{1, ..., L\} \text{ and } k \in \{1, ..., K\} \tag{3}$$

We optimize our critic network with the following objective:

$$\sum_k \sum_l \left( Q_\theta^{l,k}(\mathbf{h}_t, \mathbf{a}_{t+k-1}^l, \mathbf{a}_{t:t+K}^{l-1}) - r_{t+1} - \gamma \max_{a'} Q_{\bar{\theta}}^{l,k}(\mathbf{h}_{t+1}, a', \pi_K^l(\mathbf{h}_{t+1})) \right)^2, \tag{4}$$

where $\pi_K^l$ is an action sequence policy that outputs the action sequence $\mathbf{a}_{t:t+K}^l$. In practice, we compute Q-values for all sequence step $k \in \{1, ..., K\}$ in parallel, which is possible because Q-values for future actions depend only on current features $\mathbf{h}_t$ but not on Q-values for previous actions. We find this simple design, with independence across action sequence, works well even on challenging humanoid control tasks with high-dimensional action spaces (Sferrazza et al., 2024). We expect our idea can be strengthened by exploiting the sequential structure, *i.e.,* Q-values at subsequent steps depend on previous Q-values (Metz et al., 2017; Chebotar et al., 2023), but we leave it as future work.

**Architecture**   We implement our critic network to initially extract features for each sequence step $k$ and aggregate features from multiple steps with a recurrent network (see Figure 2). This architecture is often helpful in cases where a single-step action is already high-dimensional so that concatenating them make inputs too high-dimensional. Specifically, let $\mathbf{e}_k$ denote an one-hot encoding for $k$. At each level $l$, we construct features for each sequence step $k$ as $\mathbf{h}_{t,k}^l = \left[ \mathbf{h}_t, \mathbf{a}_{t+k-1}^{l-1}, \mathbf{e}_k \right]$. We then encode each $\mathbf{h}_{t,k}^l$ with a shared MLP network and process them through GRU (Cho et al., 2014) to obtain $\mathbf{s}_{t,k}^l = f_\theta^{\texttt{GRU}}(f_\theta^{\texttt{MLP}}(\mathbf{h}_{t,1}^l), ..., f_\theta^{\texttt{MLP}}(\mathbf{h}_{t,k}^l))$. We then use a shared projection layer to map each $\mathbf{s}_{t,k}^l$ into Q-values at each sequence step $k$, *i.e.,* $Q_\theta^{l,k}(\mathbf{o}_t, \mathbf{a}_{t+k-1}^l, \mathbf{a}_{t:t+K}^{l-1}) = f_\theta^{\texttt{proj}}(\mathbf{s}_{t,k}^l)$.

## 3.2 Action Execution and Training Details

While the idea of using action sequence is simple, there are two important yet small details for effectively training RL agents with action sequence: (i) how we execute actions at each time step to control robots and (ii) how we store training data and sample batches for training.

**Executing action with temporal ensemble**   With the policy that outputs an action sequence $\mathbf{a}_{t:t+K}$, one important question is how to execute actions at time step $i \in \{t, ..., t+K-1\}$. For this, we use the idea of Zhao et al. (2023) that utilizes *temporal ensemble*, which computes $\mathbf{a}_{t:t+K}$ every time step, saves it to a buffer, and executes a weighted average $\sum_i w_i \mathbf{a}_{t-i} / \sum w_i$ where $w_i = \exp(-m * i)$ denotes a weight that assigns higher value to more recent actions. We find this scheme outperforms the alternative of computing $\mathbf{a}_{t:t+K}$ every $K$ steps and executing each action for subsequent $K$ steps on most tasks we considered, except on several tasks that need reactive control.

**Storing training data from environment interaction**   When storing samples from online environment interaction, we store a transition $(\mathbf{o}_t, \hat{\mathbf{a}}_t, r_{t+1}, \mathbf{o}_{t+1})$ where $\hat{\mathbf{a}}_t$ denotes an action executed at time step $t$. For instance, if we use temporal ensemble for action execution, $\hat{\mathbf{a}}_t$ is a weighted average of action outputs obtained from previous $K$ time steps.

**Sampling training data from a replay buffer**   When sampling training data from the replay buffer, we sample a transition with action sequence, *i.e.,* $(\mathbf{o}_t, \hat{\mathbf{a}}_{t:t+K}, r_{t+1}, \mathbf{o}_{t+1})$. If we sample time step $t$ near the end of episode so that we do not have enough data to construct a full action sequence, we fill the action sequence with *null* actions. In particular, in position control where we specify the position of joints or end effectors, we repeat the action from the last step so that the agent learns not to change the position. On the other hand, in torque control where we specify the force to apply to joints, we set the action after the last step to zero so that agent learns to not to apply force.

Figure 3: **Examples of robotic tasks.** We study CQN-**AS** on 53 robotic tasks spanning mobile bi-manual manipulation, whole-body control, and tabletop manipulation tasks from BiGym (Chernyadev et al., 2024), HumanoidBench (Sferrazza et al., 2024), and RLBench (James et al., 2020).

## 4 EXPERIMENT

We study CQN-**AS** on 53 robotic tasks spanning mobile bi-manual manipulation, whole-body control, and tabletop manipulation tasks from BiGym (Chernyadev et al., 2024), HumanoidBench (Sferrazza et al., 2024), and RLBench (James et al., 2020) environments (see Figure 3 for examples of robotic tasks). These tasks with sparse and dense rewards, with or without vision sensors, and with or without demonstrations, allow for evaluating the capabilities and limitations of our algorithm. In particular, our experiments are designed to investigate the following questions:

- Can CQN-**AS** quickly match the performance of a recent BC algorithm (Zhao et al., 2023) and surpass it through online learning? How does CQN-**AS** compare to previous model-free RL algorithms (Haarnoja et al., 2018; Yarats et al., 2022; Seo et al., 2024)?
- What is the contribution of each component in CQN-**AS**?
- Under which conditions is CQN-**AS** effective? When does CQN-**AS** fail?

**Baselines for fine-grained control tasks with demonstrations**  For tasks that need high-precision control, *e.g.,* manipulation tasks from BiGym and RLBench, we consider model-free RL baselines that aim to learn deterministic policies, as we find that stochastic policies struggle to solve such fine-grained control tasks. Specifically, we consider (i) Coarse-to-fine Q-Network (CQN; Seo et al. 2024), a value-based RL algorithm that learns to zoom-into continuous action space in a coarse-to-fine manner, and (ii) DrQ-v2+, an optimized demo-driven variant of a model-free actor-critic algorithm DrQ-v2 (Yarats et al., 2022) that uses a deterministic policy algorithm and data augmentation. We further consider (iii) Action Chunking Transformer (ACT; Zhao et al. 2023), a BC algorithm that trains a transformer (Vaswani et al., 2017) policy to predict action sequence and utilizes temporal ensemble for executing actions, as our highly-optimized BC baseline.

**Baselines for whole-body control tasks with dense reward**  For locomotion tasks with dense reward, we consider (i) Soft Actor-Critic (SAC; Haarnoja et al. 2018), a model-free actor-critic RL algorithm that maximizes action entropy, and (ii) Coarse-to-fine Q-Network (CQN; Seo et al. 2024). Moreover, although it is **not** the goal of this paper to compare against model-based RL algorithms, we also consider two model-based baselines: (iii) DreamerV3 (Hafner et al., 2023), a model-based RL algorithm that learns a latent dynamics model and a policy using imagined trajectories and (iv) TD-MPC2 (Hansen et al., 2024), a model-based RL algorithm that learns a latent dynamics model and utilizes local trajectory optimization in imagined latent trajectories.

**Implementation details**  For training with expert demonstrations, we follow the setup of Seo et al. (2024). Specifically, we keep a separate replay buffer that stores demonstrations and sample half of training data from demonstrations. We also relabel successful online episodes as demonstrations and store them in the demonstration replay buffer. For CQN-**AS**, we use an auxiliary BC loss from Seo et al. (2024) based on large margin loss (Hester et al., 2018). For actor-critic baselines, we use an auxiliary BC loss that minimizes L2 loss between the policy outputs and expert actions. We implement CQN-**AS** based on a publicly available source code of CQN[2] based on PyTorch (Paszke et al., 2019). We will release the full source code upon publication.

---

[2]https://github.com/younggyoseo/CQN

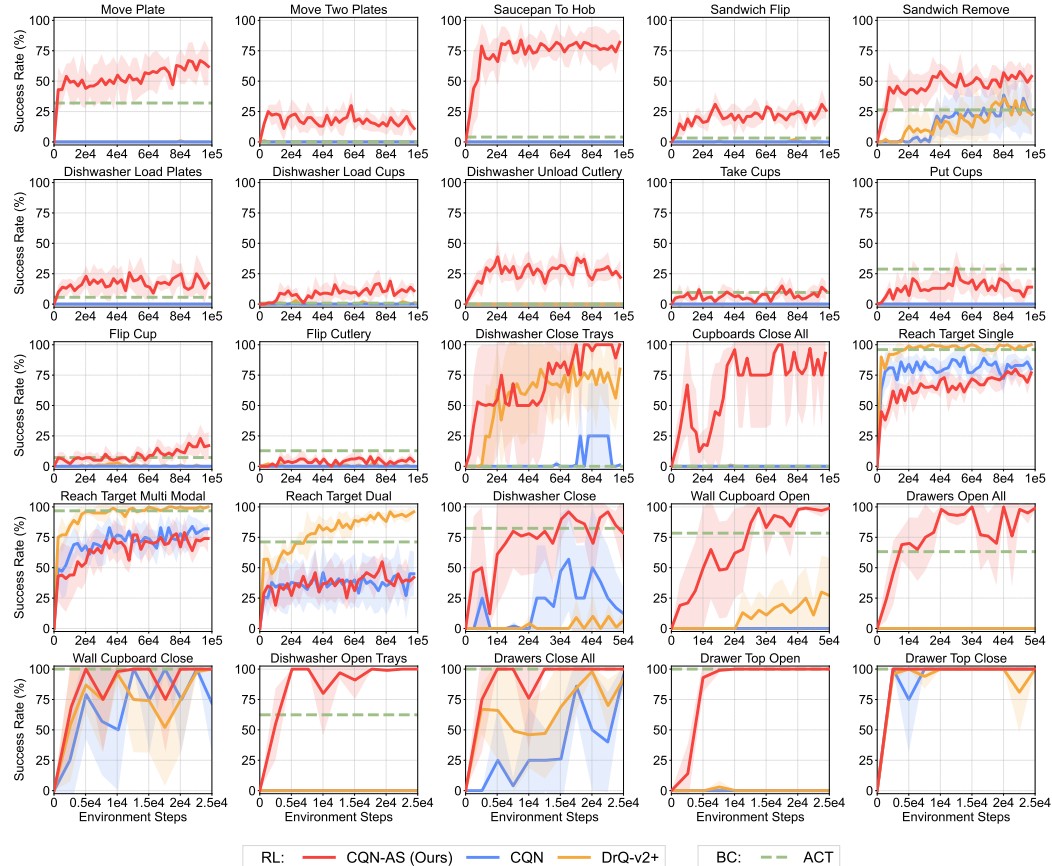

Figure 4: **BiGym results** on 25 sparsely-rewarded mobile bi-manual manipulation tasks (Chernyadev et al., 2024). All experiments are initialized with 17 to 60 *human-collected* demonstrations, and RL methods are trained with an auxiliary BC objective. On many of the challenging long-horizon tasks, CQN-**AS** quickly learns to match the performance of ACT (Zhao et al., 2023) and surpass it through online learning, while other RL baselines fail to effectively accelerate training with noisy human-collected demonstrations. We report the success rate over 25 episodes. The solid line and shaded regions represent the mean and confidence intervals, respectively, across 4 runs.

## 4.1 BiGym Experiments

We study CQN-**AS** on mobile bi-manual manipulation tasks from BiGym (Chernyadev et al., 2024). BiGym's *human-collected* demonstrations are often noisy and multi-modal, posing challenges to RL algorithms which should effectively leverage demonstrations for solving sparsely-rewarded tasks.

**Setup** Because we find that not all demonstrations from BiGym benchmark can be successfully replayed[3], we replay all the demonstrations and only use the successful ones as demonstrations. We do not discard ones that fail to be replayed, but we use them as training data with zero reward because they can still be useful as failure experiences. To avoid training with too few demonstrations, we exclude the tasks where the ratio of successful demonstrations is below 50%. This leaves us with 25 tasks, each with 17 to 60 demonstrations. For visual observations, we use RGB observations with 84×84 resolution from head, left_wrist, and right_wrist cameras. We also use low-dimensional proprioceptive states from proprioception, proprioception_grippers, and proprioception_floating_base sensors. We use (i) absolute joint position control action mode and (ii) floating base that replaces locomotion with classic controllers. We use the same set of hyperparameters for all the tasks, in particular, we use action sequence of length 16. More details on BiGym experiments are available in Appendix A.

---

[3]We use demonstrations available at the date of Oct 1st with the commit 018f8b2.

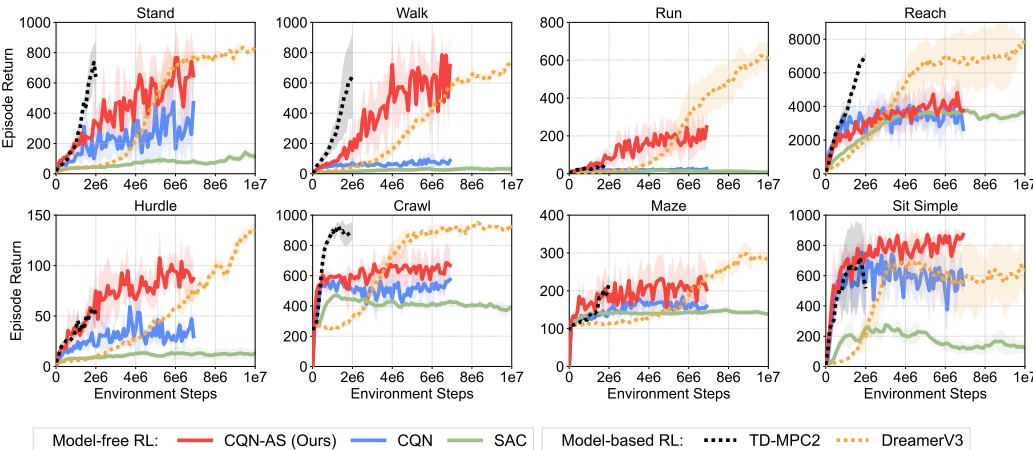

Figure 5: **HumanoidBench results** on 8 densely-rewarded humanoid control tasks (Sferrazza et al., 2024). All the experiments start from scratch and RL methods do not have an auxiliary BC objective. CQN-**AS** significantly outperforms other model-free RL baselines on most tasks. CQN-**AS** often achieves competitive performance to model-based RL baselines, which is intriguing but not the main goal of this paper. For CQN-**AS** and CQN, we report the results aggregated over 4 runs. For other baselines, we report the results aggregated over 3 runs available from public website. The solid line and shaded regions represent the mean and confidence intervals.

**Comparison to baselines**   Figure 4 shows the experimental results on BiGym benchmark. We find that CQN-**AS** quickly matches the performance of ACT and outperforms it through online learning on 20/25 tasks, while other RL algorithms fail to do so especially on challenging long-horizon tasks such as Move Plate and Saucepan To Hob. A notable result here is that CQN-**AS** *enables* solving these challenging BiGym tasks while other RL baselines, *i.e.,* CQN and DrQ-v2+, completely fail as they achieve 0% success rate. This result highlights the capability of CQN-**AS** to accelerate RL training from noisy, multi-modal demonstrations collected by humans.

**Limitation**   However, we find that CQN-**AS** struggles to achieve meaningful success rate on some of the long-horizon tasks that require interaction with delicate objects such as cup or cutlery, leaving room for future work to incorporate advanced vision encoders (He et al., 2016; 2022) or critic architectures (Kapturowski et al., 2023; Chebotar et al., 2023; Springenberg et al., 2024).

### 4.2   HUMANOIDBENCH EXPERIMENTS

To show that CQN-**AS** can be generally applicable to tasks without demonstrations, we study CQN-**AS** on densely-rewarded humanoid control tasks from HumanoidBench (Sferrazza et al., 2024).

**Setup**   For HumanoidBench, we follow a standard setup that trains RL agents from scratch, which is also used in original benchmark (Sferrazza et al., 2024). Specifically, we use low-dimensional states consisting of proprioception and privileged task information as inputs. For tasks, we simply select the first 8 locomotion tasks in the benchmark. Following the original benchmark that trains RL agents for environment steps that roughly requires 48 hours of training, we report the results of CQN-**AS** and CQN for 7 million steps. For baselines, we use the results available from the public repository, which are evaluated on tasks with dexterous hands, and we also evaluate our algorithm on tasks with hands. We use the same set of hyperparameters for all the tasks, in particular, we use action sequence of length 4. More details on HumanoidBench experiments are available in Appendix A.

**Comparison to model-free RL baselines**   Figure 5 shows the results on on HumanoidBench. We find that, by learning the critic network with action sequence, CQN-**AS** outperforms other model-free RL baselines, *i.e.,* CQN and SAC, on most tasks. In particular, the difference between CQN-**AS** and baselines becomes larger as the task gets more difficult, *e.g.,* baselines fail to achieve high episode return on Walk and Run tasks but CQN-**AS** achieves strong performance. This result shows that our idea of using action sequence can be applicable to generic setup without demonstrations.

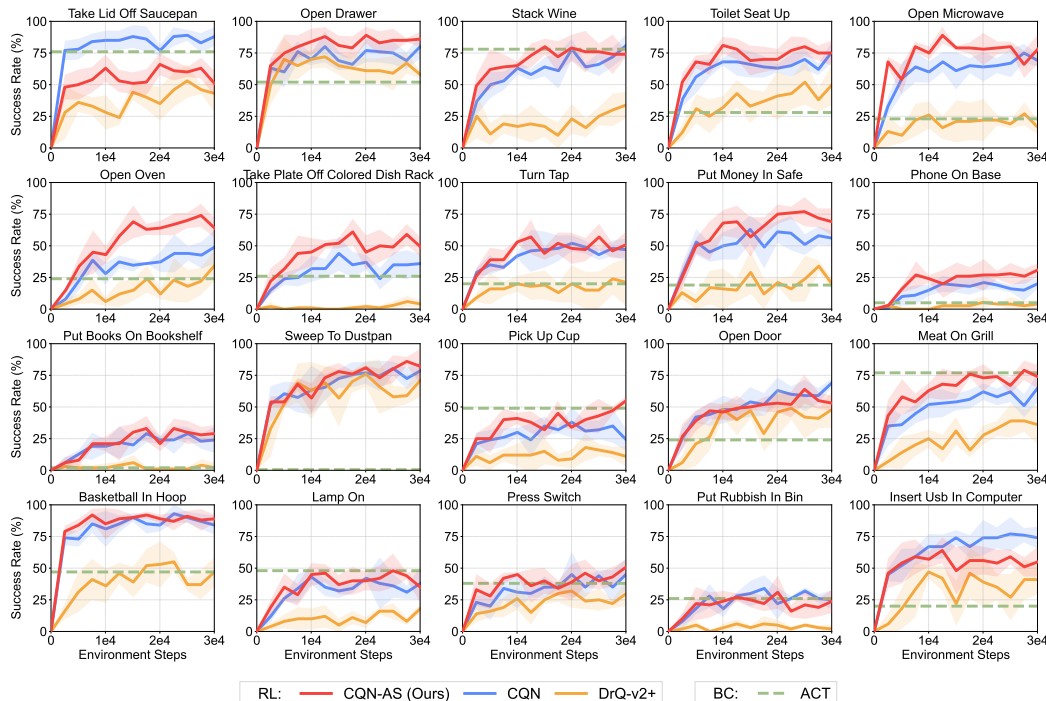

Figure 6: **RLBench results** on 20 sparsely-rewarded tabletop manipulation tasks from RLBench (James et al., 2020). All experiments are initialized with 100 *synthetic* demonstrations generated via motion-planning and RL methods are trained with an auxiliary BC objective. *As expected*, with synthetic demonstrations, CQN-**AS** achieves *similar* performance to CQN on most tasks. However, CQN-**AS** often significantly outperforms baselines on several challenging, long-horizon tasks such as Open Oven. We report the success rate over 25 episodes. The solid line and shaded regions represent the mean and confidence intervals, respectively, across 4 runs.

**CQN-AS often achieves competitive performance to model-based RL baselines** While outperforming model-based RL algorithms is not the goal of this paper, we find that CQN-**AS** often achieves competitive performance to model-based RL baselines, *i.e.,* DreamerV3 and TD-MPC2, on tasks such as Run or Sit Simple. This result shows the potential of our idea to enable RL agents to learn useful value functions on challenging tasks, without the need to explicitly learn dynamics model. We also note that incorporating our idea into world model learning could be an interesting direction.

### 4.3 RLBENCH EXPERIMENTS

To investigate whether CQN-**AS** can also be effective in leveraging *clean* demonstrations, we study CQN-**AS** on RLBench (James et al., 2020) with synthetic demonstrations.

**Setup** For RLBench experiments, we use the official CQN implementation for collecting demonstrations and reproducing the baseline results on the same set of tasks. Specifically, we use RGB observations with 84×84 resolution from front, wrist, left_shoulder, and right_shoulder cameras. We also use low-dimensional proprioceptive states consisting of 7-dimensional joint positions and a binary value for gripper open. We use 100 demonstrations and delta joint position control action mode. We use the same set of hyperparameters for all the tasks, in particular, we use action sequence of length 4. More details on RLBench experiments are available in Appendix A.

**CQN-AS is also effective with *clean* demonstrations** Because RLBench provides synthetic *clean* demonstrations, as we expected, Figure 6 shows that CQN-**AS** achieves *similar* performance to CQN on many of the tasks, except 2/25 tasks where it hurts the performance. But we still find that CQN-**AS** achieves quite superior performance to CQN on some challenging long-horizon tasks such as Open Oven or Take Plate Off Colored Dish Rack. These results, along with results from BiGym and HumanoidBench, show that CQN-**AS** can be used in various benchmark with different characteristics.

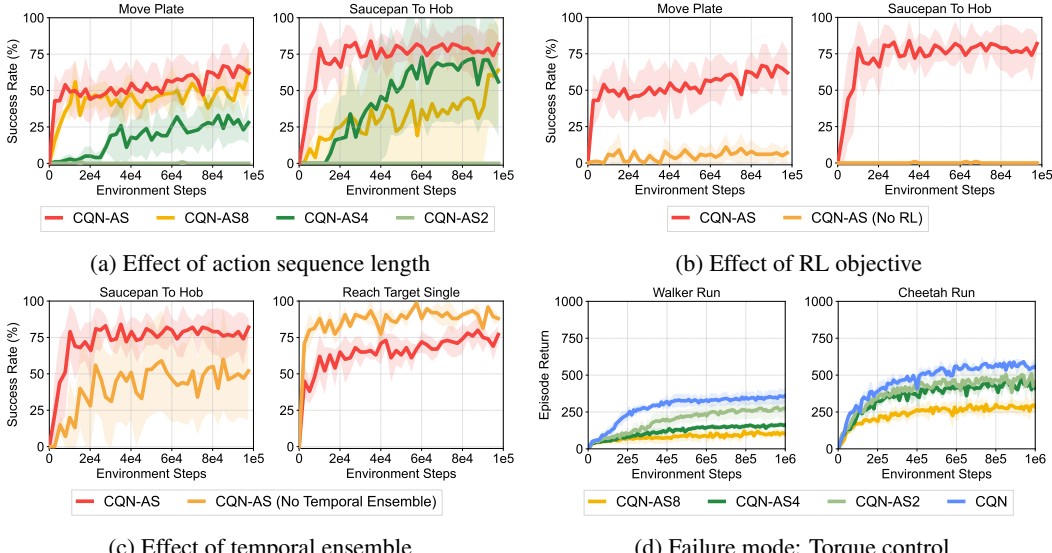

(a) Effect of action sequence length         (b) Effect of RL objective

(c) Effect of temporal ensemble         (d) Failure mode: Torque control

Figure 7: **Ablation studies and analysis** on the effect of (a) action sequence, (b) RL objective, and (c) temporal ensemble. (d) We also provide results on locomotion tasks from DeepMind Control Suite (Tassa et al., 2020), where CQN-**AS** fails to improve performance. The solid line and shaded regions represent the mean and confidence intervals, respectively, across 4 runs.

## 4.4 ABLATION STUDIES, ANALYSIS, FAILURE CASES

**Effect of action sequence length**    Figure 7a shows the performance of CQN-**AS** with different action sequence lengths on two BiGym tasks. We find that training the critic network with longer action sequences improves and stabilizes performance.

**RL objective is crucial for strong performance**    Figure 7b shows the performance of CQN-**AS** without RL objective that trains the model only with BC objective on successful demonstrations. We find this baseline significantly underperforms CQN-**AS**, which shows that RL objective is indeed enabling the agent to learn from online trial-and-error experiences.

**Effect of temporal ensemble**    Figure 7c shows that performance largely degrades without temporal ensemble on Saucepan To Hop as temporal ensemble induces a smooth motion and thus improves performance in fine-grained control tasks. But we also find that temporal ensemble can be harmful on Reach Target Single. We hypothesize this is because temporal ensemble often makes it difficult to refine behaviors based on recent visual observations. Nonetheless, we use temporal ensemble for all the tasks as we find it helps on most tasks and we aim to use the same set of hyperparameters.

**Failure case: Torque control**    Figure 7d shows that CQN-**AS** underperforms CQN on locomotion tasks with torque control. We hypothesize this is because a sequence of joint positions usually has a semantic meaning in joint spaces, making it easier to learn with, when compared to learning how to apply a sequence of torques. Addressing this failure case is an interesting future direction.

## 5 RELATED WORK

**Behavior cloning with action sequence**    Recent behavior cloning approaches have shown that predicting a sequence of actions enables the policy to effectively imitate noisy expert trajectories and helps in dealing with idle actions from human pauses during data collection (Zhao et al., 2023; Chi et al., 2023). In particular, Zhao et al. (2023) train a transformer model (Vaswani et al., 2017) that predicts action sequence and Chi et al. (2023) train a denoising diffusion model (Ho et al., 2020) that approximates the action distributions. This idea has been extended to multi-task setup (Bharadhwaj et al., 2024), mobile manipulation (Fu et al., 2024b) and humanoid control (Fu et al., 2024a). Our work is inspired by this line of work and proposed to learn RL agents with action sequence.

**Reinforcement learning with action sequence** In the context of reinforcement learning, Medini & Shrivastava (2019) proposed to pre-compute frequent action sequences from expert demonstrations and augment the action space with these sequences. However, this idea introduces additional complexity and is not scalable to setups without demonstrations. One recent work relevant to ours is Saanum et al. (2024) that encourages a sequence of actions from RL agents to be predictable and smooth. But this differs from our work in that it uses the concept of action sequence only for computing the penalty term. Recently, Ankile et al. (2024) point out that RL with action sequence is challenging and instead proposes to use RL for learning a single-step policy that corrects action sequence predictions from BC. In contrast, our work shows that training RL agents with action sequence is feasible and leads to improved performance compared to prior RL algorithms.

## 6 CONCLUSION

We presented Coarse-to-fine Q-Network with **A**ction **S**equence (CQN-**AS**), a value-based RL algorithm that trains a critic network that outputs Q-values over action sequences. Extensive experiments in benchmarks with various setups show that our idea not only improves the performance of the base algorithm but also allows for solving complex tasks where prior RL algorithms completely fail.

We believe our work will be strong evidence that shows RL can realize its promise to develop robots that can continually improve through online trial-and-error experiences, surpassing the performance of BC approaches. We are excited about future directions, including real-world RL with humanoid robots, incorporating advanced critic architectures (Kapturowski et al., 2023; Chebotar et al., 2023; Springenberg et al., 2024), bootstrapping RL agents from imitation learning (Hu et al., 2023; Xing et al., 2024) or offline RL (Nair et al., 2020; Lee et al., 2021), extending the idea to recent model-based RL approaches (Hafner et al., 2023; Hansen et al., 2024), to name but a few.

## REPRODUCIBILITY STATEMENT

We have provided details required to implement our algorithm and reproduce the results in Section 4 and Appendix A. We will release the full source code upon publication.

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

## A  EXPERIMENTAL DETAILS

**BiGym**    BiGym[4] (Chernyadev et al., 2024) is built upon MuJoCo (Todorov et al., 2012). We use Unitree H1 with two parallel grippers. We find that demonstrations available in the recent version of BiGym are not all successful. Therefore we adopt the strategy of replaying all the demonstrations and only use the successful ones as demonstrations. instead of discarding the failed demonstrations, we still store them in a replay buffer as failure experiences. To avoid training with too few demonstrations, we exclude the tasks where the ratio of successful demonstrations is below 50%. Table 1 shows the list of 25 sparsely-rewarded mobile bi-manual manipulation tasks used in our experiments.

Table 1: **BiGym tasks** with their maximum episode length and number of successful demonstrations.

| Task | Length | Demos | Task | Length | Demos |
|------|--------|-------|------|--------|-------|
| Move Plate | 300 | 51 | Cupboards Close All | 620 | 53 |
| Move Two Plates | 550 | 30 | Reach Target Single | 100 | 30 |
| Saucepan To Hob | 440 | 28 | Reach Target Multi Modal | 100 | 60 |
| Sandwich Flip | 620 | 34 | Reach Target Dual | 100 | 50 |
| Sandwich Remove | 540 | 24 | Dishwasher Close | 375 | 44 |
| Dishwasher Load Plates | 560 | 17 | Wall Cupboard Open | 300 | 44 |
| Dishwasher Load Cups | 750 | 58 | Drawers Open All | 480 | 45 |
| Dishwasher Unload Cutlery | 620 | 29 | Wall Cupboard Close | 300 | 60 |
| Take Cups | 420 | 32 | Dishwasher Open Trays | 380 | 57 |
| Put Cups | 425 | 43 | Drawers Close All | 200 | 59 |
| Flip Cup | 550 | 45 | Drawer Top Open | 200 | 40 |
| Flip Cutlery | 500 | 43 | Drawer Top Close | 120 | 51 |
| Dishwasher Close Trays | 320 | 62 | | | |

**HumanoidBench**    HumanoidBench[5] (Sferrazza et al., 2024) is built upon MuJoCo (Todorov et al., 2012). We use Unitree H1 with two dexterous hands. We consider the first 8 locomotion tasks in the benchmark: Stand, Walk, Run, Reach, Hurdle, Crawl, Maze, Sit Simple. We use proprioceptive states and privileged task information instead of visual observations. Unlike BiGym and RLBench experiments, we do not utilize dueling network (Wang et al., 2016) and distributional critic (Bellemare et al., 2017) in HumanoidBench for faster experimentation.

**RLBench**    RLBench[6] (James et al., 2020) is built upon CoppeliaSim (Rohmer et al., 2013) and PyRep (James et al., 2019). We use a 7-DoF Franka Panda robot arm and a parallel gripper. Following the setup of Seo et al. (2024), we increase the velocity and acceleration of the arm by 2 times. For all experiments, we use 100 demonstrations generated via motion-planning. Table 2 shows the list of 20 sparsely-rewarded visual manipulation tasks used in our experiments.

Table 2: **RLBench tasks** with their maximum episode length used in our experiments.

| Task | Length | Task | Length |
|------|--------|------|--------|
| Take Lid Off Saucepan | 100 | Put Books On Bookshelf | 175 |
| Open Drawer | 100 | Sweep To Dustpan | 100 |
| Stack Wine | 150 | Pick Up Cup | 100 |
| Toilet Seat Up | 150 | Open Door | 125 |
| Open Microwave | 125 | Meat On Grill | 150 |
| Open Oven | 225 | Basketball In Hoop | 125 |
| Take Plate Off Colored Dish Rack | 150 | Lamp On | 100 |
| Turn Tap | 125 | Press Switch | 100 |
| Put Money In Safe | 150 | Put Rubbish In Bin | 150 |
| Phone on Base | 175 | Insert Usb In Computer | 100 |

---

[4]https://github.com/chernyadev/bigym

[5]https://github.com/carlosferrazza/humanoid-bench

[6]https://github.com/stepjam/RLBench

**Hyperparameters** We use the same set of hyperparameters across the tasks in each domain. For hyperparameters shared across CQN and CQN-**AS**, we use the same hyperparameters for both algorithms for a fair comparison. We provide detailed hyperparameters for BiGym and RLBench experiments in Table 3 and hyperparameters for HumanoidBench experiments in Table 4

Table 3: Hyperparameters for demo-driven vision-based experiments in BiGym and RLBench

| Hyperparameter | Value |
|---|---|
| Image resolution | $84 \times 84 \times 3$ |
| Image augmentation | RandomShift (Yarats et al., 2022) |
| Frame stack | 4 (BiGym) / 8 (RLBench) |
| CNN - Architecture | Conv (c=[32, 64, 128, 256], s=2, p=1) |
| MLP - Architecture | Linear (c=[512, 512, 64, 512, 512], bias=False) (BiGym) |
| | Linear (c=[64, 512, 512], bias=False) (RLBench) |
| CNN & MLP - Activation | SiLU (Hendrycks & Gimpel, 2016) and LayerNorm (Ba et al., 2016) |
| GRU - Architecture | GRU (c=[512], bidirectional=False) |
| Dueling network | True |
| C51 - Atoms | 51 |
| C51 - $v_{min}$, $v_{max}$ | -2, 2 |
| Action sequence | 16 (BiGym) / 4 (RLBench) |
| Temporal ensemble weight $m$ | 0.01 |
| Levels | 3 |
| Bins | 5 |
| BC loss ($\mathcal{L}_{BC}$) scale | 1.0 |
| RL loss ($\mathcal{L}_{RL}$) scale | 0.1 |
| Relabeling as demonstrations | True |
| Data-driven action scaling | True |
| Action mode | Absolute Joint (BiGym), Delta Joint (RLBench) |
| Exploration noise | $\epsilon \sim \mathcal{N}(0, 0.01)$ |
| Target critic update ratio ($\tau$) | 0.02 |
| N-step return | 1 |
| Batch size | 128 (BiGym) / 256 (RLBench) |
| Demo batch size | 128 (BiGym) / 256 (RLBench) |
| Optimizer | AdamW (Loshchilov & Hutter, 2019) |
| Learning rate | 5e-5 |
| Weight decay | 0.1 |

Table 4: Hyperparameters for state-based experiments in HumanoidBench

| Hyperparameter | Value |
|---|---|
| MLP - Architecture | Linear (c=[512, 512], bias=False) |
| CNN & MLP - Activation | SiLU (Hendrycks & Gimpel, 2016) and LayerNorm (Ba et al., 2016) |
| GRU - Architecture | GRU (c=[512], bidirectional=False) |
| Dueling network | False |
| Action sequence | 4 |
| Temporal ensemble weight $m$ | 0.01 |
| Levels | 3 |
| Bins | 5 |
| RL loss ($\mathcal{L}_{RL}$) scale | 1.0 |
| Action mode | Absolute Joint |
| Exploration noise | $\epsilon \sim \mathcal{N}(0, 0.01)$ |
| Target critic update ratio ($\tau$) | 1.0 |
| Target critic update interval ($\tau$) | 1000 |
| N-step return | 3 |
| Batch size | 128 |
| Optimizer | AdamW (Loshchilov & Hutter, 2019) |
| Learning rate | 5e-5 |
| Weight decay | 0.1 |

**Computing hardware** For all experiments, we use consumer-grade 11GB GPUs such as NVIDIA GTX 1080Ti, NVIDIA Titan XP, and NVIDIA RTX 2080Ti with 11 or 12GB VRAM. With 2080Ti GPU, each BiGym experiment with 100K environment steps take 9.5 hours, each RLBench experiment with 30K environment steps take 6.5 hours, and each HumanoidBench experiment with 7M environment steps take 48 hours. We find that CQN-**AS** is around 33% slower than running CQN because larger architecture slows down both training and inference.

**Baseline implementation** For CQN (Seo et al., 2024) and DrQ-v2+ (Yarats et al., 2022), we use the implementation available from the official CQN implementation[7]. For ACT (Zhao et al., 2023), we use the implementation from RoboBase repository[8]. For SAC (Haarnoja et al., 2018), DreamerV3 (Hafner et al., 2023), and TD-MPC2 (Hansen et al., 2024), we use results provided in HumanoidBench[9] repository (Sferrazza et al., 2024).

## B FULL DESCRIPTION OF CQN AND CQN-AS

This section provides the formulation of CQN and CQN-**AS** with $n$-dimensional actions.

### B.1 COARSE-TO-FINE Q-NETWORK

Let $a_t^{l,n}$ be an action at level $l$ and dimension $n$ and $\mathbf{a}_t^l = \{a_t^{l,1}, ..., a_t^{l,N}\}$ be actions at level $l$ with $\mathbf{a}_t^0$ being zero vector. We then define coarse-to-fine critic to consist of multiple Q-networks:

$$Q_\theta^{l,n}(\mathbf{h}_t, a_t^{l,n}, \mathbf{a}_t^{l-1}) \text{ for } l \in \{1, ..., L\} \text{ and } n \in \{1, ..., N\} \tag{5}$$

We optimize the critic network with the following objective:

$$\sum_n \sum_l \left( Q_\theta^{l,n}(\mathbf{h}_t, a_t^{l,n}, \mathbf{a}_t^{l-1}) - r_{t+1} - \gamma \max_{a'} Q_{\bar{\theta}}^{l,n}(\mathbf{h}_{t+1}, a', \pi^l(\mathbf{h}_{t+1})) \right)^2, \tag{6}$$

where $\bar{\theta}$ are delayed parameters for a target network (Polyak & Juditsky, 1992) and $\pi^l$ is a policy that outputs the action $\mathbf{a}_t^l$ at each level $l$ via the inference steps with our critic, *i.e.*, $\pi^l(\mathbf{h}_t) = \mathbf{a}_t^l$.

**Action inference** To output actions at time step $t$ with the critic, CQN first initializes constants $a_t^{n,\texttt{low}}$ and $a_t^{n,\texttt{high}}$ with $-1$ and $1$ for each $n$. Then the following steps are repeated for $l \in \{1, ..., L\}$:

- Step 1 (Discretization): Discretize an interval $[a_t^{n,\texttt{low}}, a_t^{n,\texttt{high}}]$ into $B$ uniform intervals, and each of these intervals become an action space for $Q_\theta^{l,n}$

- Step 2 (Bin selection): Find the bin with the highest Q-value, set $a_t^{l,n}$ to the centroid of the selected bin, and aggregate actions from all dimensions to $\mathbf{a}_t^l$

- Step 3 (Zoom-in): Set $a_t^{n,\texttt{low}}$ and $a_t^{n,\texttt{high}}$ to the minimum and maximum of the selected bin, which intuitively can be seen as zooming-into each bin.

We then use the last level's action $\mathbf{a}_t^L$ as the action at time step $t$.

**Computing Q-values** To compute Q-values for given actions $\mathbf{a}_t$, CQN first initializes constants $a_t^{n,\texttt{low}}$ and $a_t^{n,\texttt{high}}$ with $-1$ and $1$ for each $n$. We then repeat the following steps for $l \in \{1, ..., L\}$:

- Step 1 (Discretization): Discretize an interval $[a_t^{n,\texttt{low}}, a_t^{n,\texttt{high}}]$ into $B$ uniform intervals, and each of these intervals become an action space for $Q_\theta^{l,n}$

- Step 2 (Bin selection): Find the bin that contains input action $\mathbf{a}_t$, compute $a_t^{l,n}$ for the selected interval, and compute Q-values $Q_\theta^{l,n}(\mathbf{h}_t, a_t^{l,n}, \mathbf{a}_t^{l-1})$.

- Step 3 (Zoom-in): Set $a_t^{n,\texttt{low}}$ and $a_t^{n,\texttt{high}}$ to the minimum and maximum of the selected bin, which intuitively can be seen as zooming-into each bin.

We then use a set of Q-values $\{Q_\theta^{l,n}(\mathbf{h}_t, a_t^{l,n}, \mathbf{a}_t^{l-1})\}_{l=1}^L$ for given actions $\mathbf{a}_t$.

---

[7] https://github.com/younggyoseo/CQN
[8] https://github.com/robobase-org/robobase
[9] https://github.com/carlosferrazza/humanoid-bench

## B.2 COARSE-TO-FINE CRITIC WITH ACTION SEQUENCE

Let $\mathbf{a}_{t:t+K}^l = \{\mathbf{a}_t^l, ..., \mathbf{a}_{t+K-1}^l\}$ be an action sequence at level $l$ and $\mathbf{a}_{t:t+K}^0$ be zero vector. Our critic network consists of multiple Q-networks for each level $l$, dimension $n$, and sequence step $k$:

$$Q_\theta^{l,n,k}(\mathbf{h}_t, a_{t+k-1}^{l,n}, \mathbf{a}_{t:t+K}^{l-1}) \text{ for } l \in \{1, ..., L\}, \ n \in \{1, ..., N\} \text{ and } k \in \{1, ..., K\} \qquad (7)$$

We optimize the critic network with the following objective:

$$\sum_n \sum_l \sum_k \left( Q_\theta^{l,n,k}(\mathbf{h}_t, a_t^{l,n}, \mathbf{a}_{t:t+K}^{l-1}) - r_{t+1} - \gamma \max_{a'} Q_{\bar{\theta}}^{l,n,k}(\mathbf{h}_{t+1}, a', \pi_K^l(\mathbf{h}_{t+1})) \right)^2, \qquad (8)$$

where $\pi_K^l$ is an action sequence policy that outputs the action sequence $\mathbf{a}_{t:t+K}^l$. In practice, we compute Q-values for all sequence step $k \in \{1, ..., K\}$ and all action dimension $n \in \{1, ..., N\}$ in parallel. This can be seen as extending the idea of Seyde et al. (2023), which learns decentralized Q-networks for action dimensions, into action sequence dimension. As we mentioned in Section 3.1, we find this simple scheme works well on challenging tasks with high-dimensional action spaces.

**Architecture**  Let $\mathbf{e}_k$ denote an one-hot encoding for $k$. For each level $l$, we construct features for each sequence step $k$ as $\mathbf{h}_{t,k}^l = \left[\mathbf{h}_t, \mathbf{a}_{t+k-1}^{l-1}, \mathbf{e}_k\right]$. We encode each $\mathbf{h}_{t,k}^l$ with a shared MLP network and process them through GRU (Cho et al., 2014) to obtain $\mathbf{s}_{t,k}^l = f_\theta^{\text{GRU}}(f_\theta^{\text{MLP}}(\mathbf{h}_{t,1}^l), ..., f_\theta^{\text{MLP}}(\mathbf{h}_{t,k}^l))$. We use a shared projection layer to map each $\mathbf{s}_{t,k}^l$ into Q-values at each sequence step $k$, *i.e.,* $\{Q_\theta^{l,k}(\mathbf{o}_t, a_{t+k-1}^{l,n}, \mathbf{a}_{t:t+K}^{l-1})\}_{n=1}^N = f_\theta^{\text{proj}}(\mathbf{s}_{t,k}^l)$. We note that we compute Q-values for all dimensions $n \in \{1, ..., N\}$ at the same time with a big linear layer, which follows the design of Seo et al. (2024).

