# OpenReview forum: "Reinforcement Learning with Action Sequence for Data-Efficient Robot Learning"
_ICLR.cc/2025/Conference — Submitted to ICLR 2025_

### Official Review · Reviewer_u3LC · 2024-10-30

**Soundness:** 2
**Presentation:** 3
**Contribution:** 3
**Rating:** 5
**Confidence:** 4

**Summary:**

The paper proposes an extension to the Coarse-to-fine Q-Network (CQN) reinforcement learning algorithm that predicts Q-values for sequences of actions. Using sequences of actions instead of single actions is inspired by work on behaviroal cloning that demonstrated that predicting sequences of actions improves the model's ability to fit noisy and multi-modal distributions of expert demonstrations. The authors evaluate their method on simulated benchmark tasks from BiGym, HumanoidBench, and RLBench, where the method demonstrates performance gains compared to the original CQN method, as well as SAC and DrQ-v2+.

**Strengths:**

The paper is easy to follow and the modifications to the original CQN algorithm are explained clearly. The evaluation is conducted on a wide variety of simulated robotics tasks. The authors evaluate both the pure RL setting as well as RL with demonstrations and include ablations for key components of the algorithm.

**Weaknesses:**

The method focuses only on the CQN algorithm while the concept of predicting Q-values for action sequences rather than individual actions should be applicable to other reinforcement learning methods as well. It remains unclear whether such a modification would also lead to similar performance gains in other RL algorithms.

The method appears to train a large number of Q-functions (L * K, where L = 3 and K = 16 for some of the experiments). This large number of neural networks certainly reduces the computational efficiency of the method. Some comparison of the computational efficiency would be nice.

**Questions:**

1. I am unsure why giving the Q-function access to the action sequence is beneficial if the Q-function is still trained with just a one-step Bellman error. This way the Q-function is not really getting a learning signal for the rest of the sequence. For example, a valid solution to the loss from equation 4 could be to ignore all actions but $a_t$ and predict the Q-value only from $o_t$ and $a_t$, like a regular Q-function without access to the action sequence. Since this is the central component of the paper, I would appreciate greatly if the authors could expand on why the action sequence is beneficial if the Q-functions are trained with this loss and how the Q-functions are incentivized to make use of the entire sequence for the predictions.

2. There seems to be an ablation for the temporal ensemble missing. While the authors compare averaging the actions predicted at different timesteps and executing the entire action sequence, the authors do not show the results of predicting the actions at every step and always executing only the first action. The current "No temporal ensemble" ablation essentially reduces the control frequency of the task, which certainly can have a negative impact on the agent's performance, but this does not necessarily mean that it is beneficial to average actions predicted at multiple timesteps.

3. In some of the plots the methods are not trained for enough steps. E.g., in the Run plot of Figure 5, TD-MPC2 is trained for only 2e6 steps and at this point it seems to be on-par with CQN-AS, but it is unclear whether it will perform better or worse in the long run.

4. The plots in Figure 1 are lacking standard deviations, which makes it hard to assess the significance of the results. Furthermore, all experiments are repeated for only 4 runs. It would be good increase the number of runs since the performance of RL algorithms can be a quite stochastic.

5. According to equation 2, every Q-function only sees the bin that was chosen at the current level and one level before. The authors use the method with three levels (according to Table 3). How would the finest Q-function know which bin was chosen at the coarsest level? Clearly, the bin chosen at the coarsest level has a strong impact on actual action executed by the agent and consequently on the Q-values. If the method would learn a deterministic policy in an on-policy setting, the agent would always choose the same action for a given state and the finer-level Q-function would need to adapt to the changed meaning of the actions only when the agent is updated. However, the method is able to process data from expert demonstrations and therefore needs to be able to process off-policy data, so it is unclear to me how the Q-function at the finest level is able to infer which bin was chosen at the highest level. I am aware that this is rather a question about the original CQN method than CQN-AS, but since understanding CQN is essential for understanding the method, I would appreciate if the authors could clarify this issue.

6. In and around equations 2 and 4, some of the actions are bold, and some (e.g., lines 132, 141) are not. I was wondering what the difference between the bold and non-bold actions is.

7. What exactly is $\pi^l_K$ in equation 4? Is it a learned policy or does it just denote the maximization over the Q-functions?

8. Line 187: Why is the one-hot vector $e_k$ necessary? If I understand the architecture correctly, the GRU gets the $h^l_{t,k}$ in sequence, so it could just learn to count to get the k. Why is k nevertheless needed as an explicit input?

9. According to lines 187-189 $s^l_{t,k}$ contains no information about $a^{l}$, only $a^{l-1}$ and in line 191 the Q-function is defined as a mapping that takes only $s^l_{t,k}$ as input, but the Q-function is supposed to also have access to $a^{l}$. Where is $a^l$ added to the inputs?

10. Line 201: m is not defined. Is it a hyperparameter?

11. Line 201: The notation $a_{t-i}$ is a bit confusing. Previously $a_{t-i}$ would mean the action for timestep t-i, but here it is supposed to mean the action for timestep t calculated at timestep t-i, I think. I think the author should make that explicit.

12. Lines 203/204 state that computing the actions only every K steps and applying the K-step action sequence is beneficial for tasks that require reactive control. This is a bit puzzling to me since executing the action sequence open-loop should make the policy less reactive since the agent can only react every K steps, while in the temporal ensemble case, the agent can at least change the actions at every step to some extent.

13. Line 302: The claim "other RL baselines fail to effectively accelerate training with noisy human-collected demonstrations" is a bit misleading. It sounds as if the baselines perform on-par with CQN-AS if all methods are trained without demonstrations, and CQN-AS excels when given demonstrations, but Figure 4 does not contain experiments without demonstrations. It would be good to clarify that statement.

14. Line 405: The authors describe the "Open Oven" task as a "challenging, long-horizon task". While I have not worked with RLBench yet, I cannot imagine why the "Open Oven" task would have a longer horizon tasks than other tasks (e.g., "Open Door").

15. The claim "we find that CQN-AS often achieves competitive performance to model-based RL baselines, i.e., DreamerV3 and TD-MPC2, on tasks such as Run or Sit Simple" does not seem to be backed by the data for the "Run" task. On this task DreamerV3 achieves significantly higher performance and the TD-MPC2 run was stopped too early to allow for a proper comparison.


Typos:
1. Line 264: "of training" --> "of the training"

2. Line 377: "generic setup" --> "generic setups"

---

> ### Author Response · Authors · 2024-11-24
> **Response to Reviewer u3LC [1/3]**
>
> Dear Reviewer u3LC,
>
> We sincerely thank you for your feedback and insightful comments to improve the manuscript. We address your comment below.
>
> ---
>
> **[W1]  It remains unclear whether such a modification would also lead to similar performance gains in other RL algorithms.**
>
> **[A-W1]** Thank you for pointing this out. We have included the [results of DrQ-v2-**AS**+](https://anonymous.4open.science/r/cqn_as_rebuttal_anonymous-5B35/drqv2_as_plus.png) which trains both critic and actor with action sequences. This significantly outperforms DrQ-v2+ in the BiGym environment, demonstrating that our idea is applicable to other RL algorithms. We will include more results in the final draft.
>
> ---
>
> **[W2] Training multiple Q-networks for all the levels and action sequence will be slow. Some comparison of the computational efficiency would be nice.**
>
> **[A-W2]** Thank you for the suggestion. In practice, we do not train all Q-networks separately but train a single network that shares most of the parameters by following the design of [1]. Specifically, the network can share most of the parameters by taking level index and step index as inputs. Nonetheless, it makes training slower as you pointed out, i.e., CQN-**AS** is approximately 33\% slower than CQN as we mentioned in ‘Computing hardware’ of Appendix A. We will further clarify this in the final draft.
>
> [1] Seyde, Tim, et al. "Solving continuous control via q-learning." ICLR 2023
>
> ---
>
> **[Q1] Is Q-function getting a learning signal for all actions within the action sequence? Wouldn’t Q-function ignore all actions other than $a_{t}$?**
>
> **[A1]** We’d like to first clarify that our Q-network does not take $a_{t:t+K}$ as inputs, but it outputs a $K$-dimensional vector for all $K$ actions within the action sequence, i.e., $ (Q(o_t, a_t), Q(o_t, a_{t+1}), \ldots, Q(o_t, a_{t+K-1})) $. Therefore Q-function does not ignore other actions. More specifically, because we explicitly train $Q(o_{t}, a_{t+k})$ to predict $r\_{t+1} + \gamma Q(o\_{t+1}, a’\_{t+k})$, where $a’\_{t+k}$ is the action that maximizes Q-values of action index $k$, for all $k$, each Q-network can learn the effect of outputting action $a\_{t+k}$ at the current timestep. We will further clarify this in the final draft.
>
> ---
>
> **[Q2] Additional ablation on CQN-AS that computes actions at each timestep and uses the first action**
>
> **A2.** Thank you for your suggestion. We have included [the results of CQN-**AS** (First Action)](https://anonymous.4open.science/r/cqn_as_rebuttal_anonymous-5B35/ablation_first_action.png) that computes actions at every timestep but always uses the first action in the sequence. We find that only using the first action performs significantly worse than both CQN-**AS** and CQN-**AS** (No Temporal Ensemble). This shows that a mismatch between the training time (train a model to learn Q-values for a sequence of actions) and the test time (only use the first action from the output action sequence) is harmful for the performance. We will include the relevant results in the final draft.
>
> ---
>
> **[Q3] In some of the plots the methods are not trained for enough steps. E.g., in the Run plot of Figure 5, TD-MPC2 is trained for only 2e6 steps and at this point it seems to be on-par with CQN-AS, but it is unclear whether it will perform better or worse in the long run.**
>
> **[A3]** We’d like to emphasize that comparison against model-based RL methods is not our main focus. For this reason, we used the numbers in the [official HumanoidBench repository](https://github.com/carlosferrazza/humanoid-bench) without running model-based RL methods by ourselves. But we agree that it will be informative for follow-up research, so we will add the results in the final draft.
>
> ---
>
> **[Q4] Missing standard deviations in Figure 1. Increasing the number of runs from 4.**
>
> **[A4]** Thank you for pointing this out. We have included the [aggregate plot that includes the confidence interval](https://anonymous.4open.science/r/cqn_as_rebuttal_anonymous-5B35/figure1_with_ci.png). For increasing the number of runs, We could not include results with more runs because it is very costly to run experiments on 53 tasks considered in our experiments. We will increase the number of runs in the final draft.
>
> ---
>
> **[Q5] How does the Q-network know which bin was chosen at the coarsest level?**
>
> **[A5]** CQN uses the numerical values corresponding to the center of each bin as inputs to the next level’s Q-network, so the network has the information required to understand where it is located within the interval. But it would be interesting to investigate if giving previous actions from all the levels as inputs make any difference. We will clarify this in the final draft.

---

> ### Author Response · Authors · 2024-11-24
> **Response to Reviewer u3LC [2/3]**
>
> **[Q6] In and around equations 2 and 4, some of the actions are bold, and some (e.g., lines 132, 141) are not. I was wondering what the difference between the bold and non-bold actions is.**
>
> **[A6]** Thank you for pointing this out. As we mentioned in line 829 of Appendix B, non-bold action corresponds to action at each dimension, and bold-action denotes the action vector consisting of actions from all dimensions. We will update this in the final draft.
>
> ---
>
> **[Q7] What exactly is $\pi_{K}^{l}$ in Equation 4? Is it a learned policy or does it just denote the maximization over the Q-functions?**
>
> **[A7]** It denotes the policy that outputs action via action inference which we introduced in line 138-144 of the original draft. We will clarify this.
>
> ---
>
> **[Q8] Why is the one-hot vector $e_{k}$ necessary?**
>
> **[A8]** Thank you for a good question. $e_{k}$ is necessary because we’re first extracting features with the MLP network $f_{\theta}^{\texttt{MLP}}$ and then using GRU network to aggregate features. Without $e_{k}$, the MLP would not know where this $\mathbf{a}_{t+k-1}^{l-1}$ comes from. While the training signal from GRU would allow for training MLP networks without explicit index, we find that this empirically works better.
>
> ---
>
> **[Q9] Where is $a^{l}$ added to the inputs?**
>
> **[A9]** We would like to clarify that our network follows the design of value-based RL algorithms such as DQN, therefore the Q-network doesn’t take $a^{l}$ as inputs, but it outputs Q-values for each $a^{l}$.
>
> ---
>
> **[Q10] Line 201: $m$ is not defined. Is it a hyperparameter?**
>
> **[A10]** Thank you for pointing this out. $m$ is a hyperparameter used for aggregating the actions in the temporal ensemble, which was originally introduced in [2] as a hyperparameter $k$. We did use $m=0.01$ following [1].
>
> [2] Zhao, Tony Z., et al. "Learning fine-grained bimanual manipulation with low-cost hardware." RSS 2023
>
> ---
>
> **[Q11] Modify notation $a_{t-i}$ in Line 201 to explicitly denote that this is action calculated at timestep $t-i$**
>
> **[A11]** Thank you for your suggestion. We will make it explicit by denoting it as $\bar{a}_{t}^{i}$ and mentioning that it is an action for timestep $t$ which is calculated at timestep $t-i$. Please let us know if you have any better suggestions.
>
> ---
>
> **[Q12] Lines 203/204 state that computing the actions only every K steps and applying the K-step action sequence is beneficial for tasks that require reactive control. This is a bit puzzling to me since executing the action sequence open-loop should make the policy less reactive since the agent can only react every K steps, while in the temporal ensemble case, the agent can at least change the actions at every step to some extent.**
>
> **[A12]** This is a great question. You’re correct in pointing out that policy with temporal ensemble can be more reactive in that it can compute actions every time step. However, in practice, we find that temporal ensemble makes the policy *less reactive* because previous actions from $K-1$ steps affect the action at the current time step. In contrast, without a temporal ensemble, the policy can directly execute the most recent actions predicted by the network, without being affected by the predictions from the previous timesteps. We will add more discussion on this point, thank you for pointing this out.

---

> > ### Author Response · Authors · 2024-11-24
> > **Response to Reviewer u3LC [3/3]**
> >
> > **[Q13] Line 302: The claim "other RL baselines fail to effectively accelerate training with noisy human-collected demonstrations" is a bit misleading. It sounds as if the baselines perform on-par with CQN-AS if all methods are trained without demonstrations, and CQN-AS excels when given demonstrations, but Figure 4 does not contain experiments without demonstrations. It would be good to clarify that statement.**
> >
> > **[A13]** We agree with you in that it can be confusing. We will simply state that other RL baselines fail to achieve competitive performance in BiGym.
> >
> > ---
> >
> > **[Q14] Line 405: The authors describe the "Open Oven" task as a "challenging, long-horizon task". While I have not worked with RLBench yet, I cannot imagine why the "Open Oven" task would have a longer horizon tasks than other tasks (e.g., "Open Door").**
> >
> > **[A14]** This is an interesting question. Open Oven is challenging because of the characteristic of an oven itself and also 7-DOF Franka Arm. Because the oven is *thicker* than objects like a door in RLBench, Franka arm must make a wide circular movement from its initial position to grasp the oven handle. Also, because of the joint limit of Franka, the robot cannot open the oven in a single pull. Instead it has to first pull the handle and then push the oven door further down. This makes the Open Oven task longer-horizon and more challenging compared to other tasks.
> >
> > ---
> >
> > **[Q15] The claim "we find that CQN-AS often achieves competitive performance to model-based RL baselines, i.e., DreamerV3 and TD-MPC2, on tasks such as Run or Sit Simple" does not seem to be backed by the data for the "Run" task. On this task DreamerV3 achieves significantly higher performance and the TD-MPC2 run was stopped too early to allow for a proper comparison.**
> >
> > **[A15]** We would like to emphasize that we stated CQN-AS **often** achieves competitive performance to model-based RL algorithms. Because the comparison with model-based RL is not our main focus, we did not intend to claim that our method is consistently outperforming model-based RL algorithms. We will further clarify this by emphasizing that it is not our main focus. We will also state that CQN-AS **sometimes** achieves competitive performance.

---

> ### Comment · Reviewer_u3LC · 2024-11-25
>
> [A-W1]:
> Thank you for these additional experiments. These first results look very promising.
>
> [A-W2]:
> I see, so the Q-network essentially looks like $Q(h_t, l, k) = q_{t+k}^l$ where $q_{t+k}^l$ is a DQN-style vector of Q-values for all actions on the level $l$ for the timestep $t+k$, predicted at timestep $t$? Please consider adding to the manuscript that the Q-functions are parameterized as a single network with the index input and that the network outputs vectors of Q-values. In my opinion, this is not clear from the manuscript.
>
> [A1]:
> I still do not understand this part. $a_{t+k}$ is the action that is (potentially) executed at $k$ steps into the future. It is updated with the reward for the current step $r_{t+1}$, but $a_{t+k}$ should not have an influence on $r_{t+1}$ (or $h_{t+1}$), right? So, couldn't $Q^k$ just predict the value $Q(s_t, a_t)$ for all outputs, i.e., the regular one-step Q-value, ignoring $a_{t+k}$? I think that should be a valid solution to equation 4.
>
> I feel like the loss should be something along the lines of $(Q^k(h_t, a_{t+k-1}^l, a_{t:t+K}^{l-1}) - \sum_{\tau=t}^{t+k} r_\tau - max_{a'} Q^k(h_{t+k}, a', \pi_K^l(h_{t+k})))^2$. Could you please expand on how exactly the loss from equation 4 works?
>
> [A2]:
> Thank you for this additional ablation. I am a bit puzzled that this setting does not work at all. I see that there is a mismatch between training on sequences of actions and then testing with single actions, but still I feel that the first action of the sequence should still do something at least somewhat meaningful. Could you expand on what you think are the reasons why this setting does not learn anything meaningful at all for these tasks?
>
> [A3]:
> Yes, for the final version, it would be very interesting to see the comparison between CQN-AS and the model-based baselines when trained for the same number of steps.
>
> [A4]:
> Thank you for adding the confidence intervals. Regarding the number of seeds, I can see that computation capacities can be a limiting factor here.
>
> [A5]:
> I see, so the $a_t^{l-1}$ passed to Q-function $Q^l$ is not a representation of "bin x on level l", but rather the mid-point of the bin chosen on level l, represented in the original action space?
>
> [A6]:
> This is a bit confusing since the footnote on page 3 states that all actions are assumed to be one-dimensional in the main paper.
>
> [A7]:
> Thanks for the clarification.
>
> [A8]:
> I see, so the network could learn to count, but it is easier to simply add the index as an additional input. Please consider adding the explanation to the paper as it could make this part a bit clearer.
>
> [A9]:
> Thanks for the clarification. As mentioned in [A-W1], I think it would be beneficial for the sake of clarity to mention this in the paper.
>
> [A10]:
> Thank you for clarifying this.
>
> [A11]:
> Yes, that would make this clearer.
>
> [A12]:
> Okay, so let's say K=5, and something that the policy needs to react to happens at timestep 3, this would mean that the "no temporal ensemble" policy would get full control over the actions at timestep 5, while the "temporal ensemble" policy would still be influenced by the previous actions at that point. Only at timestep 8 would this policy be free of the influence of the previous actions. Is this intuition correct? If I understand correctly, the hyperparameter m essentially controls how reactive the "temporal ensemble" policy is. Did you try different values for m to see if that makes the approach more suited for tasks that require reactive control?
>
> [A13]:
> Yes, changing the statement in that way would make it more accurate.
>
> [A14]:
> I see, that makes sense. Thanks for the explanation.
>
> [A15]:
> My main concern here was that you used the "Run" task as an example where CQN-AS achieves competitive performance to the model-based RL baselines, which does not seem to be the case according to Figure 5. So please do not use this task as an example here. I am happy with the updated statement.

---

> > ### Author Response · Authors · 2024-12-02
> >
> > Dear Reviewer u3LC,
> >
> > We deeply appreciate your feedback on the draft. Here are the responses to your additional questions and comments. We promise to incorporate all the comments you left for improving our draft.
> >
> > ---
> > **On the DQN-style architecture**
> >
> > **[A-W2]** Thank you for the suggestion, we’ll update the manuscript as you suggested.
> >
> > ---
> > **On the Equation 4 that uses a single-step reward**
> >
> > **[A1]** In Equation 4, we agree that using N-step return can be more intuitive. But we would like to note that each Q-network knows which time step it is dealing with and Q-value is trained with bootstrapping. For instance, if we think about a sparse-reward setup with an episode length of $H$, the model can know that action for $H+k$ is an action after the episode succeeds, and can propagate that information to previous steps through bootstrapping.
> >
> > ---
> > **On the results that show using the first action does not work at all**
> >
> > **[A2]** Thank you for having a detailed look at the additional result. We hypothesize the reason why using the first action did not work is that the tasks we considered in the rebuttal response were challenging tasks, so the mismatch significantly degrades the performance. In our follow-up experiments, we find that using the first action performs reasonably well on simple tasks such as Reach Target Single, achieving similar performance to CQN, a baseline that does not use action sequence.
> >
> > ---
> > **On the details of previous actions as inputs**
> >
> > **[A5]** Your understanding is correct. They are not representations of the bins but actual scalar values in the original action space.
> >
> > ---
> > **On the notation of bold and non-bold actions**
> >
> > **[A6]** We will clarify this in the final draft.
> >
> > ---
> > **On the effect of hyperparameter $m$ in temporal ensemble**
> >
> > **[A12]** Your understanding is correct, and yes, $m$ does indeed control how reactive the temporal ensemble policy can be. We did not try tuning the value but it could be interesting to see how it affects the performance on tasks with different characteristics.

---

> > > ### Comment · Reviewer_u3LC · 2024-12-03
> > >
> > > [A-W2]: Thank you, I believe that would make it clearer.
> > >
> > > [A1]: Did you at some point try N-step returns? Would that also work with your approach?
> > >
> > > With the current loss, I can see that the Q-function is able to observe that the agent succeeds k steps into the future, but I do not see how the loss incentivizes the Q-function to make use of that knowledge. Bootstrapping with single-step rewards should only enforce single-step consistency of the Q-values, not k-step consistency, right? Would the example from my previous response (where the Q-function outputs the "regular" Q-value $Q(s_t, a_t)$ for all $a_{t+k}$) be a valid minimum of the loss or am I missing something here?
> > >
> > > [A2]: It is quite interesting that this strategy works for simple tasks, thanks for this additional insight. However, it still remains a bit puzzling to me why it does not work at all on the harder tasks. After all, to get sequence prediction right, it is also necessary to do a decent job on the first prediction. Did you observe any erratic behavior of the "first action" policy? Something like switching between different strategies with high frequency, and therefore not achieving the goal? I could imagine that the temporal ensemble could be able to smooth out such problems.
> > >
> > > [A5]: Thank you for the clarification.
> > >
> > > [A6]: Thank you, that would make the equations easier to understand.
> > >
> > > [A12]: I assume that this hyperparameter has a lot of influence on the performance for certain tasks, so such experiments could yield important insights into how suitable the method is for reactive control.

---

### Official Review · Reviewer_xAUB · 2024-11-03

**Soundness:** 3
**Presentation:** 2
**Contribution:** 2
**Rating:** 5
**Confidence:** 4

**Summary:**

The paper extends the existing Coarse-to-fine Q-Network (Seo et al. 2024) by predicting Q value over a sequence of actions. The intuition behind this design is that it may learn a more robust Q function for noisy demonstrations. The authors conducted a thorough comparison against many baselines on various types of control problems, including BiGym, HumanoidBench, and RLBench.

**Strengths:**

* The paper presents a simple-yet-effective idea that can improve the performance of CQN algorithm.
* The proposed method seems to be effective on various control problems, including manipulation and humanoid control, which is impressive.

**Weaknesses:**

* I am not convinced that Q-prediction with action sequences offers higher accuracy over noisy trajectories. I believe this is the most important claim in the paper, however, I cannot find a good discussion. The paper simply mentions that "our algorithm allows for learning useful value functions from noisy trajectories." However, this is not intuitive; I am not sure why adding more arguments to the Q network mitigates such problems. Examples and/or toy problems would be appreciated.
* In that sense, I am not sure how the proposed technique is generic to other RL algorithms. Is it only specific to CQN? If I combine a similar idea with other off-the-shelf algorithms, such as PPO or SAC, would it also offer a significant performance boost?
* The paper is less self-contained. The paper suddenly uses the term "level", which is never properly introduced in this paper. Therefore, I believe the current manuscript forces readers to visit the previous CQN paper, which is not desirable.
* The paper uses both RL and BC as problems, and I am not so sure it is a good idea. The paper may focus on one of the problems to consolidate the scenario.

**Questions:**

Please refer to the comments above.

---

> ### Author Response · Authors · 2024-11-24
> **Response to Reviewer xAUB [1/1]**
>
> Dear Reviewer xAUB,
>
> We sincerely thank you for feedback and insightful comments to improve the draft. We address your comment below.
>
> ---
>
> **[Q1-2] Intuition on why learning Q-network with action sequence is effective on domains with noisy trajectories. Is the proposed technique generic to other RL algorithms?**
>
> **[A1-2]** We have included the [results of DrQ-v2-**AS**+](https://anonymous.4open.science/r/cqn_as_rebuttal_anonymous-5B35/drqv2_as_plus.png), which trains both critic and actor with action sequences. This significantly outperforms DrQ-v2+ in the BiGym environment, which shows that our idea of learning RL agents with action sequence is applicable to other RL algorithms. We will include more results in the final draft.
>
> To provide a more intuition on why our idea is helpful, we would like to illustrate an example where a human collects a demonstration of a robot moving forward. Due to noise in the data collection system or human error, the robot may jitter, moving left-forward or right-forward at each timestep, but ultimately proceeds straight. In this case, prior approaches train a Q-network to learn the effect of executing each action $a_{t}$ at the current timestep, but it could be challenging because each action is *noisy* in a sense that it is not going forward. On the other hand, our approach explicitly trains the Q-value to learn the effect of a series of actions that ultimately move forward, which makes it easier for the Q-network to understand the effect of executing actions. We will include a relevant discussion on the final draft.
>
> ---
>
> **[Q3] The paper is less self-contained. The paper suddenly uses the term "level", which is never properly introduced in this paper. Therefore, I believe the current manuscript forces readers to visit the previous CQN paper, which is not desirable.**
>
> **[A3]** We agree that some of the definitions are missing and Figure 2 is not fully self-contained. While we provided a detailed background of CQN in the original draft’s Section 2 in an effort to make the paper as self-contained as possible, we will further complement our Background section to make the paper more self-contained. Thank you for pointing this out.
>
> ---
>
> **[Q4] The paper uses both RL and BC as problems, and I am not so sure it is a good idea. The paper may focus on one of the problems to consolidate the scenario.**
>
> **[A4]** We believe that empirical evaluation on diverse setups with or without demonstrations is one of the main strengths of our paper, as the other two reviewers also highlighted in their reviews. Considering the recent advances in data collection systems and interest in using expert demonstration data for robot learning [1,2,3], we hope that our work can help facilitate future RL research to focus on both setups.
>
> [1] Zhao, Tony Z., et al. "Learning fine-grained bimanual manipulation with low-cost hardware." RSS 2023\
> [2] Fu, Zipeng, et al. "HumanPlus: Humanoid Shadowing and Imitation from Humans." CoRL 2024\
> [3] Nakamoto, Mitsuhiko, et al. "Steering Your Generalists: Improving Robotic Foundation Models via Value Guidance." CoRL 2024

---

> > ### Comment · Reviewer_xAUB · 2024-11-25
> >
> > Thank you for your feedback: I found it helpful in gaining a better understanding of the submission. However, I still believe the proposed algorithm feels more like an add-on to the CQN paper, due to the way the paper is written, which is not clearly explained in the current paper. Additionally, I am still not convinced by the intuition, as the described issues, such as robot jitteriness, could likely be addressed using alternative methods like multi-step learning or action smoothing. Despite the detailed response, I will maintain my current stance on the paper.

---

> > > ### Author Response · Authors · 2024-12-02
> > >
> > > Dear Reviewer xAUB,
> > >
> > > We deeply appreciate your feedback on the draft. Here are the responses to your additional questions and comments. We promise to incorporate all the comments you left for improving our draft.
> > >
> > > ---
> > > **On the alternative methods such as action smoothing or multi-step learning**
> > >
> > > We would like to note that (i) action smoothing can actually be harmful in fine-grained tasks as it can reduce the precision by smoothing the actions and (ii) using multi-step learning methods is orthogonal to our direction. But we agree that comparing against these baselines can further strengthen the understanding of our method, which we will incorporate in the final draft.
> > >
> > > ---
> > > **On the concern that this submission feels like an add-on to the CQN paper**
> > >
> > > While we could have provided a generic formulation of our main idea to use action sequences for RL in the Method section, we chose to focus on providing a detailed formulation of CQN and CQN-AS to make the paper as self-contained as possible within the page limits. However, we agree that the paper may be read as an extension to CQN. To address this, we will incorporate your comment by first providing the generic formulation and then giving a detailed formulation of specific algorithms. Thank you for the comment.

---

### Official Review · Reviewer_2SBg · 2024-11-03

**Soundness:** 3
**Presentation:** 3
**Contribution:** 3
**Rating:** 6
**Confidence:** 4

**Summary:**

This paper combines ideas from RL and Behavior Cloning (BC) by introducing a critic network that outputs Q-values over action sequences. The main idea is that estimating Q-values over action sequences could mitigate the effect of noisy data, especially in sparse-reward environments, and improve the agent’s anticipation of the cumulative impact of decisions over time.

The proposed Coarse-to-fine Q-Network with Action Sequence (CQN-AS) outputs Q-values for a sequence of actions, which helps the RL agent plan multiple steps ahead. This is especially useful in tasks with noisy or sparse reward signals.
By decomposing Q-values into levels $l$ and sequence steps $k$, this approach enables multi-level action consideration. The use of parallel computation for all sequence steps $K \in {1,...,K}$ is also efficient.

CQN-AS leverages a combination of MLP and GRU networks for feature encoding and processing, which is effective for handling high-dimensional input data. GRUs capture temporal dependencies, enhancing sequential learning.

In traditional reinforcement learning, actual rewards from subsequent steps are typically required for updates. However, CQN-AS addresses this differently, leveraging bootstrapping and multi-step returns to estimate future rewards without needing actual observed rewards for each step in the sequence. This allows the model to estimate returns for a series of actions based on current Q-value predictions.

**Strengths:**

- **Sequence-Based Q-Value Predictions**: Using Q-value predictions over action sequences is relatively novel afaik, particularly in reinforcement learning for robotics.
- **Strong Empirical Evaluation**: The use of tasks from BiGym, HumanoidBench, and RLBench provides a broad evaluation setup, which supports the paper’s claims of robustness.
- **Performance on Complex Robotics Tasks**: Outperforming majority of RL and BC baselines, especially in humanoid control, indicates that the method can handle complex, high-dimensional tasks.

**Weaknesses:**

- Although the approach aims to handle noise effectively, an ablation study on this can strenghten the claim. For instance, is there a mechanism to manage extreme cases of noisy trajectories, or is this approach limited in such scenarios?
- Sequential Dependency in Q-Values: The authors note that Q-values at subsequent steps are computed independently of previous Q-values, potentially overlooking dependencies across actions. While this simplifies training, it may reduce performance in tasks that require detailed planning or interdependencies between actions. Incorporating sequential dependencies between Q-values could better align with real-world robotic control needs. I'd like to know the author's thought on this.
- While the approach is empirically evaluated, there is limited theoretical analysis of why learning Q-values over action sequences should improve robustness to noise or sparse rewards. Providing some theoretical backing or intuitions for the coarse-to-fine structure could make the work more rigorous.
- If the goal is to apply this approach to real-world robotics, the paper can discuss transferability, as these are often major hurdles in robotics. Without this, it’s unclear how well CQN-AS would generalize outside simulated environments.

**Questions:**

- It can be helpful to include a brief discussion contrasting this approach with standard multi-step RL methods like $TD(\lambda)$. Traditional methods generally focus on handling delayed rewards rather than explicitly addressing noisy, multi-modal action distributions, which is a central aspect of this paper. Although the two frameworks tackle these issues differently, I believe this comparison would clarify the different approaches.
- HumanoidBench Results: The results for TD-MPC2 (Figure 5) appear to be cut off too early in the HumanoidBench experiments ? Is there a reason for this?

---

> ### Author Response · Authors · 2024-11-24
> **Response to Reviewer 2SBg [1/1]**
>
> Dear Reviewer 2SBg,
>
> We sincerely thank you for your feedback and insightful comments to improve the manuscript. We address your comment below.
>
> ---
>
> **[Q1] Although the approach aims to handle noise effectively, an ablation study on this can strenghten the claim. For instance, is there a mechanism to manage extreme cases of noisy trajectories, or is this approach limited in such scenarios?**
>
> **[A1]** Thank you for the suggestion. We believe experiments on HumanoidBench represent an extreme case for training RL agents with noisy trajectories, as the initial exploratory data consist of trajectories where humanoid robots exhibit unrealistic behaviors with jerky motions. Thus, we think that improvements on HumanoidBench demonstrate that our algorithm can effectively handle noisy trajectories. We would greatly appreciate it if the reviewer could suggest other scenarios; we would be happy to test the suggested setups.
>
> ---
>
> **[Q2] Incorporating sequential dependency between Q-values.**
>
> **[A2]** We agree with the reviewer that considering sequential dependencies, as in prior work [1], can be effective for more complex robotic tasks, as we also stated in line 182 of the original draft. However, it would introduce additional complexities requiring many techniques and details to make it work, which is why we leave it as a future work. We will include more discussion on it in the final draft.
>
> [1] Chebotar, Yevgen, et al. "Q-transformer: Scalable offline reinforcement learning via autoregressive q-functions." CoRL 2023
>
> ---
>
> **[Q3] Intuition on why action sequence works well, especially with coarse-to-fine structure**
>
> **[A3]** As we mentioned in the paper, we expect learning with action sequences makes it easy to learn from data with noisy or multi-modal distributions. However, thinking of the effect of coarse-to-fine structure is really an interesting question. We believe coarse-to-fine structure can be synergistic with learning with action sequence, because coarse-to-fine structure enables the model to quickly learn *roughly* good actions for all actions in the action sequence, and then to focus on refining actions. This may ease the difficulty of learning Q-networks with action sequences. Thank you for an interesting question, and we will include a relevant discussion in the final draft.
>
> ---
>
> **[Q4] If the goal is to apply this approach to real-world robotics, the paper can discuss transferability, as these are often major hurdles in robotics. Without this, it’s unclear how well CQN-AS would generalize outside simulated environments.**
>
> **[A4]** Thank you for pointing this out. We expect CQN-**AS** will also work well in real-world setups, as the backbone algorithm CQN already demonstrated that it can train robots to pick up objects within 10 minutes of real-world training. Moreover, considering that recent behavior cloning methods that incorporate action sequences have shown to be very effective for real-world robot control [2][3], we expect that CQN-**AS** will be also effective for controlling real robots. We will include relevant discussion in the final draft.
>
> [2] Zhao, Tony Z., et al. "Learning fine-grained bimanual manipulation with low-cost hardware." RSS 2023\
> [3] Chi, Cheng, et al. "Diffusion policy: Visuomotor policy learning via action diffusion." The International Journal of Robotics Research (2023): 02783649241273668.
>
> ---
>
> **[Q5] Discussion on the difference to multi-step RL methods**
>
> **[A5]** Thank you for the suggestion. We would like to first note that both approaches are orthogonal, so that we can incorporate multi-step RL objectives into our algorithm, as you also stated in your question. Nonetheless, the main difference of our idea to multi-step RL is that we explicitly learn the values of multiple actions inside action sequences, but multi-step RL still learns a Q-value for each action but using the rewards from multiple time steps. It would be an interesting direction to investigate the performance of the combined approach. For instance, we can consider computing the target Q-values for future actions by using the idea of multi-step RL approaches and then training Q-network with action sequence to predict such target values. We will include relevant discussion in the final draft.
>
> ---
>
> **[Q6] HumanoidBench Results: The results for TD-MPC2 (Figure 5) appear to be cut off too early in the HumanoidBench experiments ? Is there a reason for this?**
>
> **[A6]** We would like to clarify that it is not our main focus to compare CQN-**AS** with model-based RL approaches, and thus we used the numbers provided in the [official HumanoidBench repository](https://github.com/carlosferrazza/humanoid-bench) without running model-based RL methods by ourselves. We will further clarify this in the final draft.

---

> > ### Comment · Reviewer_2SBg · 2024-11-24
> > **Response to the authors**
> >
> > I acknowledge that I have read the response of the authors and other reviewers feedback and discussions. To certain extend I agree with reviewers xAUB and u3LC that there remains some issues in the paper that have not been addressed during the rebuttal. For example the key assumptions of using action sequences with one-step bellman error update, missing ablation studies, and including cut-off baseline results which were not ran so there is uncertainty how these could've performed over longer timesteps (specially in the case of TD-MPC2 baseline). I don't believe the discussion points are addressing all concerns and for that reason I will reduce my score to a 6.

---

### Meta-Review · Area_Chair_586a · 2024-12-16

**Metareview:**

This paper proposes a Q-learning based approach that predicts Q-values over sequences of actions rather than individual actions, aiming to improve data efficiency for robot learning from noisy trajectories. The main technical contribution is extending the Coarse-to-fine Q-Network (CQN) algorithm to handle action sequences, inspired by recent behavior cloning methods.

Strengths:
+ Novel application of action sequence prediction to Q-learning for robotics
+ Comprehensive evaluation on multiple robotics benchmarks (BiGym, HumanoidBench, RLBench)
+ Clear empirical improvements over baseline approaches on some tasks

Major Weaknesses:
- Fundamental questions about the core technical approach remain unaddressed, particularly how the one-step Bellman error loss incentivizes learning over action sequences
- Inadequate comparison to baselines, with some key results truncated early
- Lack of thorough ablation studies for key design choices
- Unclear theoretical motivation for why action sequence prediction should help with noisy data

**Additional Comments On Reviewer Discussion:**

After extensive discussion between reviewers and authors, significant concerns about the technical soundness and novelty remain unresolved. In particular:

- Multiple reviewers highlighted that it's unclear how single-step Bellman updates incentivize learning over action sequences, which is central to the proposed method. The authors' responses did not adequately address this core issue.
- The innovation appears limited, as noted by reviewers xAUB and dG58 - the method seems to be primarily an add-on to the existing CQN algorithm rather than a substantial advancement.
- Important empirical details are missing, like proper baselines comparisons run to completion and thorough ablation studies examining key claims.

Given these substantial unresolved issues around the technical foundations and empirical validation of the core claims, I recommend rejecting this paper. While the empirical results show promise, the lack of clear theoretical justification and thorough analysis raises too many questions about the soundness and broader applicability of the approach.

---

### Decision · Program_Chairs · 2025-01-22

Reject